# Estimation of cost efficiency of fattening pigs, sows, and piglets using SFA approach analysis: Evidence from China

Gangyi Wang[1,2], Chang'e Zhao[1]*, Yuzhuo Shen[1], Ni Yin[3]

**1** College of Economics and Management, Northeast Agricultural University, Harbin, Heilongjiang, China, **2** Chongqing University of Arts and Sciences, Chongqing, China, **3** HARBIN GuangSha College, Harbin, Heilongjiang, China

* azce@neau.edu.cn

**Data Availability Statement:** All relevant data are within the paper and its Supporting information files.

## Abstract

The hog industry is the core industry in the field of agriculture and animal husbandry in China, which development is related to the improvement of people's quality of life. The production of the hog industry has been restricted by environmental regulations, which needs to reduce costs and improve efficiency. Based on the data of 29 provinces from 2008 to 2019, this paper aims to use the stochastic frontier analysis method to calculate the cost efficiency of fattening pigs, sows, and piglets in three stages of pig production and focuses on the impact of environmental regulation policies on cost efficiency. The study found that the cost efficiency of fattening pigs, sows, and piglets in China were 0.77, 0.79, and 0.53, respectively, and the efficiency losses were 23%, 21%, and 47%, respectively. The impact of environmental regulation policies on the cost efficiency of fattening pigs showed an ' N ' trend, and the impact on the cost efficiency of sows and piglets showed an inverted ' N ' trend. For fattening pigs, increasing the intensity of environmental regulation, and exceeding the second inflection point of the ' N ' curve can achieve the dual goals of environmental governance and cost reduction and efficiency increase. For sows, reducing the intensity of environmental regulation appropriately can avoid cost-efficiency loss. For piglets, environmental regulation policies have not effectively incentives the cost efficiency of piglets. In addition, raising the level of technology investment in fattening pigs and sows can achieve cost efficiency gains, and can farmers use emerging financial product tools such as ' insurance + futures ' to avoid market risks and efficiency losses.

## Introduction

The hog industry occupies an essential position in the Chinese animal husbandry industry. As the receiver of the price of input factors, pig farmers will choose to use limited feed resources, labor resources, and land resources to increase their breeding output to achieve the goal of maximizing profits. However, due to the constraints of environmental regulations and element resources in recent years, the production costs of pig farmers have risen sharply, and the

**Funding:** This research is supported by the National Natural Science Foundation of China (grant numbers 71673273); the Natural Science Foundation of Heilongjiang Province China (grant number LH2019G002); and the Humanities and Social Science Foundation of Ministry of education of China (grant number 21YJA790053). The funders had no role in study design, data collection, and analysis, decision to publish, or preparation of the manuscript.

**Competing interests:** The authors have declared that no competing interests exist.

Chinese hog industry is at a critical stage of reducing costs and improving efficiency. The total cost of fattening pigs in 2008 was only 201.8USD/head. Following the revision of the "Environmental Protection Law" by the Standing Committee of the 12th National People's Congress in 2014, the Ministry of Agriculture promulgated the "Adjustment of the Distribution of hog industry in Southern Water Network Areas" in 2016. As a result, the environmental regulation of the hog industry has reached a new height, and the cost of fattening pigs, piglet costs, and sow prices reached the maximum in 2016, respectively 301.9USD/head, 83.8USD/head, and 258.8USD/head. Although the total cost of pigs, sows, and piglets declined slightly in 2017–2018, the overall cost is still relatively high. Faced with the constraints of resources and environment, the development of the hog industry cannot be guaranteed only by increasing the input of production factors. We need to minimize the production cost through the rational allocation of factor resources under the existing input-output situation. Farrell [1] proposed that cost efficiency reflects the effectiveness of resource allocation and utilization to a certain extent.

Research on the efficiency of livestock farming has focused on livestock farms. Scholars initially studied the efficiency of dairy farms [2, 3]. As far as we know, Sharma was the first scholar to study the production efficiency of pigs, and the results show substantial production inefficiencies among the sample swine producers [4]. Other scholars also reported similar findings (Table 1). In addition, some scholars are also concerned about the heterogeneity of operation types that currently exist in the hog industry. Tonsor and Featherstone [5] indicate considerable differences in efficiency across swine firm specializations. Po-Chi Chen [6] suggests that pigs' production phases are explicitly divided into two activities: the breed-to-farrow and wean-to-finish phases. The research shows that the sources of inefficiency in the two phases are different. Therefore, considering the heterogeneity of operation types in the hog industry, this paper estimates the cost efficiency level and influencing factors of sows, piglets, and fattening pigs from the perspective of different breeding stages.

Various studies have illustrated the different effects of personal characteristics, such as age, education, experience, and profession, or physical effects, such as farm size and specific inputs

**Table 1. Existing literature on research pig farms.**

| Year | Authors | countries & regions | Research subject | Estimated results(efficiency) |
|---|---|---|---|---|
| 1996 | Sharma K R [4], Leung P S, Zaleski H M | Hawaii | The future potential of the pig industry by determining the operational efficiency of commercial pig farmers | 0.749. |
| 1998 | Rowland W W [7], Langemeier M R, Schuler B W, et al. | Kansas, Central U.S. | This paper investigated the economic competitiveness of farrow-to-finish operations by estimating relative firm efficiency. | 0.89 |
| 2004 | Lansink AO [8], Reinhard S. | Dutch | The possibilities for improving the technical, economic, and environmental performance of Dutch pig farms | 0.90 |
| 2006 | Galanopoulos K [9], Aggelopoulos S, Kamenidou I, et al | Greece | The degree of technical and scale efficiency of commercial pig farming | 0.83 |
| 2011 | Petrovska M [10]. | The Republic of Macedonia | The level of technical efficiency on pig farms in the Republic of Macedonia | from input perspective:0.90, from output perspective: 0.87 |
| 2012 | Adetunji MO [11], Adeyemo K. | Oyo State, Nigeria | The economic efficiency of pig production. | 0.98 |
| 2013 | Latruffe L [12], Desjeux Y, Bakucs Z, et al. | Hungary | This paper investigated how production and technical efficiency would be affected if environmental regulations were fully implemented. | farrow-to-finish farms:0.55 finishing only farms: 0.423 |
| 2016 | Ly N T [13], Nanseki T, Chomei Y. | Vietnam | This paper investigated technical efficiency in household pig production and seeks to determine which factors affect it. | 0.80 |
| 2017 | Manevska-Tasevska G [14], Hansson H, Labajova K | Sweden | The influence of farm management practices on both the persistent and overall efficiency | piglet farms: 0.80, |
|  |  |  |  | finishing farms: 0.82. |

on hog efficiency estimates [8, 11, 13, 15]. With the transformation and upgrading of the hog industry, the negative externalities of the hog industry environment have become a severe problem [16]. Due to the impact of environmental laws and regulations, swine farms have incurred a 9.8% opportunity cost [17]. Tan [18] suggests that environmental regulation would significantly inhibit the hog industry. Piot-Lepetit and Moing [19] suggests that the existence of a "win-win" effect between efficiency and environmental regulation in the French pig sector. Wang et al. [20] suggests that environmental regulation has no impact on the environmental efficiency and the green total factor productivity in medium-scale pig breeding. In view of the impact of environmental regulation on pig breeding efficiency, existing studies have not reached a consistent conclusion. Therefore, this paper hopes to clarify the impact of environmental regulatory policies on the cost efficiency of the hog industry from the perspective of different breeding stages.

For the estimation of the efficiency of livestock farming, most scholars adopt two methods: Stochastic Frontier Analysis (SFA) and non-parametric Data Envelopment Analysis (DEA) [8, 9, 17, 21]. Sharma et al. [22] used DEA and SFA to measure the production efficiency in the Hawaii hog industry and compared the differences between the two measurement results. The conclusion is that the efficiency estimated by DEA was lower, and attributed this situation to the DEA approach attributes any deviation of the data points from the frontier to inefficiency. Theodoridis and Psychoudakis [23], in a study on measurement of efficiency in Greek Dairy Farm, reported similar findings. The handling of measurement errors is the crucial difference between SFA and DEA. SFA models can accommodate stochastic noise, such as measurement errors due to weather, disease, and pest infestation that is likely to be significant in farming [24]. The DEA method is susceptible to outliers since the measurement errors are ignored [25]. In empirical research, stochastic frontier analysis is widely used in the financial industry, manufacturing, and aquaculture industries (Table 2). The estimated cost efficiency of the hog industry in our study is very sensitive to random external shocks, so the paper chooses SFA to measure the estimated cost efficiency of sows, piglets, and fattening pigs.

In summary, the existing research provides analysis methods and experience references for cost efficiency estimation in this paper. This paper aims to evaluate the efficiency of fattening pigs, sows, and piglets, and focus on the impact of environmental regulation on cost efficiency. This paper hopes to provide experience for promoting the improvement of pig breeding efficiency or avoiding the loss of pig breeding efficiency under the dual constraints of resources and environment.

The rest of this paper is structured as follows. Section 2 describes the theoretical basis, Stochastic Cost Frontier function setting, and variables selection. Section 3 analyzes the spatial-temporal difference and influencing factors of cost efficiency. Section 4 concludes and proposes recommendations.

**Table 2. Literature using stochastic frontier analysis.**

| Tittle | Author | Method | Times Cited |
|---|---|---|---|
| Measuring economy-wide energy efficiency performance: A parametric frontier approach | Zhou P [26], Ang BW, Zhou DQ | SFA; DEA | 197 |
| Applying the stochastic frontier approach to measure hotel managerial efficiency in Taiwan | Chen CF [27]. | SFA | 159 |
| Exploring energy efficiency in China's iron and steel industry: A stochastic frontier approach | Boqiang Lin [28], Xiaolei Wang. | SFA | 112 |
| US residential energy demand and energy efficiency: A stochastic demand frontier approach | Filippini, M [29]. Hunt, LC. | SFA | 113 |
| Impact of land fragmentation on rice producers' technical efficiency in South-East China | Tan SH [30], N. Heerink, A. Kuyvenhoven, Qu FT | SFA | 93 |

## Materials and methods

### Theoretical framework

Farrell [1] presented the idea of the frontier function and suggested that the level of cost efficiency estimates can be calculated by the ratio of the theoretical minimum cost to actual cost. Production frontier is the production possibility boundary described by input factor set and output possibility set under specific technical level. Production frontier cost refers to the minimum cost that can be achieved in theory at a given output level [31]. The production frontier theory accurately describes the maximum possible production boundary of effective allocation of input factors, but it is difficult to empirically study and analyze the gap between the actual production allocation state and the theoretical optimal allocation state. Shephard [32, 33] proposed a distance function to combine the actual production configuration state with the production frontier that reflects the optimal production state, which provides a direct basis for measuring the actual production configuration state from the ideal production configuration state. Therefore, the production frontier theory and the distance function based on this are the theoretical basis for cost efficiency measurement.

Smith [34] proposed the division of labor theory and suggested that labor division can improve the efficiency of the market economy. Based on the labor division theory, the pig production process in many countries is divided into different phases with specialized farms devoted to sow breeding, piglet rearing, and pig fattening [35]. The division of labor theory has been fully applied in the American pig industry, and its production specialization and production efficiency are at a global leading level [36]. Tonsor and Featherstone [5] suggested that there were variations in efficiency measures both across and within operation specializations, and different efficiency enhancements should be formulated according to the specialization of production. Therefore, this paper mainly studies the cost efficiency level of sows, piglets, and fattening pigs from the perspective of different pig breeding phases, to realize the efficiency promotion or avoid efficiency inhibition of the hog industry.

In the production factor theory, capital, land, and labor are the basic production factors. However, since the proportion of land cost in each pig breeding cost is too small, this paper mainly examines the input of production factor price from the perspectives of capital and labor. The theory of cost-benefit analyzes the relationship between input and output, which is a kind of economic concept [37]. Based on the cost-benefit theory, this paper determines the specific input-output indicators.

### Model and variables setting

**Translog stochastic frontier cost model and variables setting.** To investigate the cost efficiency of sows, piglets, and fattening pigs in China, this paper establishes a stochastic cost boundary model based on Battese and Coelli [38]. The specific form of the Translog-SFA model is as follows:

$$\ln C_{hit} = \alpha_{h0} + \alpha_{h1}\ln y_{hit} + \frac{1}{2}\alpha_{h2}(\ln y_{hit})^2 + \sum_{j=1}^{3}\beta_{hj}\ln w_{hjit} + \frac{1}{2}\sum_{j=1}^{3}\sum_{k=1}^{3}\lambda_{hjk}\ln w_{hjit}\ln w_{hkit}$$

$$+ \sum_{j=1}^{3}\theta_{hj}\ln y_{hit}\ln w_{hjit} + v_{hit} + \mu_{hit} + \alpha_{h3}t + \frac{1}{2}\alpha_{h4}t^2 \qquad (1)$$

Where the subscript h represents the three stages of pig production, h = 1 represents fattening pigs, h = 2 represents sows, h = 3 represents piglets; the subscript i represents the province in the sample (i = 1, 2, . . ., n); the subscript t represents year (t = 1, 2, . . ., n); $C_{hit}$ is the total cost of h production of province i at year t. $y_{hit}$ is the output of h production of province i at year t. $\omega_{hjit}$(j = 1, 2, . . ., n) is a $j^{th}$ input price of h production of province i at year t. α, β, λ, θ are

parameters to be estimated; $v_{hit}$ is a random error term and follows $N(0, \sigma_v^2)$; $\mu_{hit}$ is the cost inefficiency item of h production of province i at year t.

According to western economics' rational economic man' hypothesis, their goal is to maximize profits. For fattening pig farmers, the slaughter weight of fattening pigs directly determines their income level, so the output of fattening pigs is measured by the slaughter weight of fattening pigs ($y_1$). For sow farmers, psy (Pigs weaned per Sow per Year) is an important indicator to measure the reproductive performance of sows, so the output of sows is measured by psy ($y_2$). For piglet farmers, raising the survival rate of fattening pigs and increasing the number of msy (Market pig per Sow per Year) are the means to maximize benefits, so the output of piglets is measured as msy ($y_3$).

So far, most scholars use capital and labor as the main input variables to measure the efficiency of pig breeding [39, 40]. The labor cost, feed cost, and piglet fee cost accounted for more than 90% of the cost of fattening pig breeding. The price changes of these three input factors directly affect the cost and benefit of fattening pig farmers. Therefore, the input factor prices of fattening pigs are mainly divided into feed price ($\omega_{11}$), piglet price ($\omega_{12}$) and labor price ($\omega_{13}$). Cao [41] suggests that the sow cost includes the purchase cost of replacement gilt, feed cost, and non-feed cost. Labor costs are a major component of non-feed costs. Therefore, the input factor price of sow mainly includes feed price ($\omega_{21}$), replacement gilt price ($\omega_{22}$) and labor price ($\omega_{23}$). Hu [42] suggests that sow cost apportioned, and feed cost accounted for about 73% of the total cost of piglets. Wang [43] suggests that labor cost and feed cost are the main factors affecting the cost of piglets. Therefore, the input factor price of piglets mainly includes feed price ($\omega_{31}$), sow apportionment price ($\omega_{32}$) and labor price ($\omega_{33}$). The theoretical framework of input and output Indicators See Fig 1.

**Inefficiency effects model and variables setting.**   Inefficiency effects are independently distributed and $\mu_{hit}$ arises by truncation (at zero) of the normal distribution with mean, $m_{hit}$, and variance, $\sigma_\mu^2$, where $\mu_{hit}$ is defined by

$$\mu_{hit} = \delta_{h0} + \sum \delta_{h1} z_{hit} + \varepsilon_{hit} \tag{2}$$

Where δ is a parameter variable to be estimated. If δ is negative, it shows that it has a negative effect on the cost inefficiency term and a positive effect on the cost efficiency, and if δ is positive, the result is the opposite. $z_{hit}$ is the intra-industry variable that affects the h production of province i at year t. Many factors are affecting the cost inefficiency of farming, such as operator experience, farm-scale, farm location, etc [44]. But this paper mainly measures the relationship between cost inefficiency index and environmental regulation, technology investment, epidemic disease, industrial structure, and other factors. Xu et al. [45] pointed out that the price

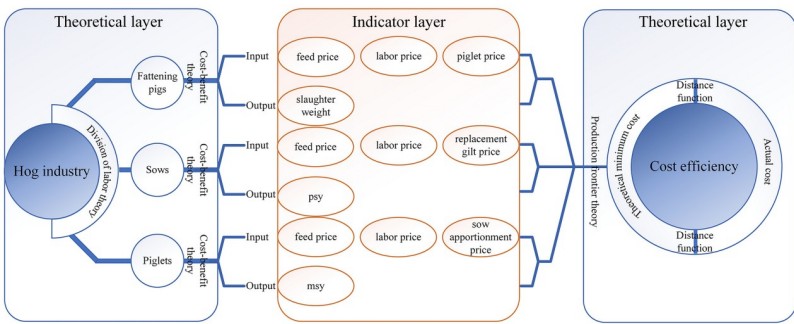

**Fig 1. Indicator selection basis.**

of corn feed had a great impact on the price of pigs, and the price of pigs was the main factor affecting the enthusiasm of farmers. When the price of pigs fell below the break-even point, farmers would reduce the number of pigs, and even be forced to slaughter sows in extreme cases. Therefore, based on considering the four influencing factors of appeal, this paper examines the effect of corn feed price on the cost inefficiency of fattening pigs, and the effect of pig price on the cost inefficiency of piglets and sows.

To sum up, this paper constructs the cost inefficiency model of fattening pigs (Formula 3) and the cost inefficiency model of sows and piglets (Formula 4), respectively. The specific form can be defined as follows:

$$\mu_{hit} = \delta_{h0} + \delta_{h1}lnenvi_{it} + \delta_{h2}(lnenvi_{it})^2 + \delta_{h3}(lnenvi_{it})^3 + \delta_{h4}tech_{it} + \delta_{h5}disea_{it} + \delta_{h6}stru_{it} + \delta_{h7}corn_{it} + \varepsilon_{hit} \tag{3}$$

$$\mu_{hit} = \delta_{h0} + \delta_{h1}lnenvi_{it} + \delta_{h2}(lnenvi_{it})^2 + \delta_{h3}(lnenvi_{it})^3 + \delta_{h4}tech_{it} + \delta_{h5}disea_{it} + \delta_{h6}stru_{it} + \delta_{h7}ppig_{it} + \varepsilon_{hit} \tag{4}$$

Where $lnenvi_{it}$ is the intensity of environmental regulation of the i province at year t. It is mainly evaluated from four aspects: policy intensity, policy objectives, policy measures, and policy feedback. Testing the nonlinear relationship between environmental regulation and cost efficiency by quadratic and cubic of environmental regulation intensity. $disea_{it}$ is the major epidemic intensity of the hog industry of the i province at year t, which is equal to the number of deaths of major diseases of fattening pigs and the number of compulsory culling divided by the total number of fattening pigs. $tech_{it}$ is the intensity of technology input of the i Province at year t, which is equal to the ratio of R & D input to GDP; $stru_{it}$ is the industrial structure of the i province at year t, which is equal to the ratio of primary industry output value to GDP. It is used to represent the agricultural resource endowment of the region. $corn_{it}$ is the corn feed price of the i province at year t to reflect the impact of factor markets prices on the efficiency of the hog industry. $ppig_{it}$ is the pig price of the i province at year t to reflect the impact of pig price on the cost efficiency of sows and piglets.

## Description of variable accounting

**The variable description of cost efficiency estimation for fattening pig breeding.**

1. Total cost ($C_1$): Expressed as the total cost of each fattening pig.

2. Output ($y_1$): Expressed by the main product output of each fattening pig, calculated according to the slaughter weight of the fattening pig.

3. Feed price ($\omega_{11}$): Expressed by dividing the cost of concentrated feed per fattening pig by the amount of concentrated feed.

4. Piglet price ($\omega_{12}$): Expressed by dividing the fee for each fattening pig by the weight of the piglet.

5. Labor price ($\omega_{13}$) Expressed by dividing the labor cost of each fattening pig by the number of laborers.

**The variable description of cost efficiency estimation for sows breeding.** (1) Total cost ($C_2$): Expressed as the total cost of each sow (Fig 2).

$$C_2 = C_{sowf} + C_{giltf} + C_{labor} \tag{5}$$

Where $C_2$ represents the total cost of sows, $C_{sowf}$ represents the cost of sows breeding feed,

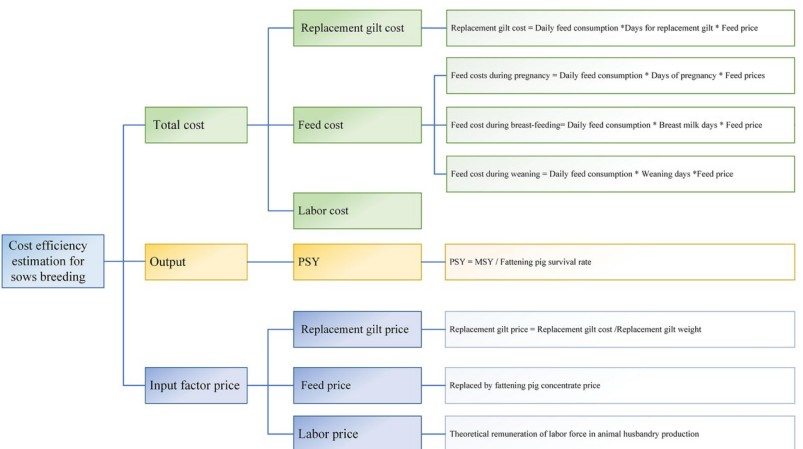

**Fig 2. Cost efficiency estimation framework for sows breeding.**

$C_{labor}$ represents the labor cost of sows raising pigs, and $C_{giltf}$ represents the cost of replacement gilt.

$$C_{sowf} = Q_{sow} \times W_{sow} \qquad (6)$$

Where $Q_{sow}$ represents the annual feed consumption of sows, and $W_{sow}$ represents the feed price of sows.

$$Q_{sow} = \left( D_{preg} \times Q_{preg} + D_{suck} \times Q_{suck} + D_{wean} \times Q_{wean} \right) \times \left[ \frac{365}{(D_{preg} + D_{suck} + D_{wean})} \right] + Q_{preg}$$
$$\times \left( D_{preg} + D_{suck} + D_{wean} \right) \times \left( \frac{365}{(D_{preg} + D_{suck} + D_{wean})} - \left[ \frac{365}{(D_{preg} + D_{suck} + D_{wean})} \right] \right) \quad (7)$$

Where $D_{preg}$ represents the pregnant days of sows of the sow, $D_{suck}$ represents the breast milk days of sows, and $D_{wean}$ represents the weaning days for sows. $Q_{preg}$ represents the daily feed consumption of the sow during pregnancy, $Q_{suck}$ represents the daily feed consumption of the sow during the breast milk period, and $Q_{wean}$ represents the daily feed consumption of the sow during the weaning period.

$$\left[ \frac{365}{(D_{preg} + D_{suck} + D_{wean})} \right]$$

$$\begin{aligned} C_{gilt} &= Q_{gilts} \times W_{gilts} \\ &= D_{gilts} \times Q_{dgilts} \times W_{gilts} \end{aligned} \qquad (8)$$

Where $Q_{gilts}$ represents the feed consumption of the replacement gilt. $W_{gilts}$ represents the feed price of the replacement gilt. $D_{gilts}$ represents the age required for the replacement gilt to reach the initial condition, and $Q_{dgilts}$ represents the daily feed consumption of the replacement gilt.

(2) Output ($y_2$): Expressed in psy per sow.

Psy refers to the number of weaned piglets that each sow can provide per year. msy is the number of fattening pigs that each sow can provide per year. The stock of fattening pigs at the end of last year is the basis for the production of fattening pigs this year, so this paper proposes

a calculation formula for calculating msy in year t:

$$msy_t = \frac{(pigso_t + pigst_t)}{\frac{sowst_{t-1} + sowst_t}{2}} \tag{9}$$

Where $pigso_t$ represents the total number of fattening pigs sold to the market in year t., $pigst_t$ represents the total number of feeding fattening pigs at the end of year t., $sowst_{t-1}$ represents the total number of sows at the end of the year $(t-1)$., and $sowst_{t-1}$ represents the total number of sows at the end of year t.

$$r_{surv} = \frac{msy_t}{psy_t} \tag{10}$$

$r_{surv}$ represents the survival rate of fattening pigs, $msy_t$ represents the number of fattening pigs that each sow can provide in year t, and $psy_t$ represents the number of weaned piglets that each sow can provide in year t.

The calculation formula of psy in year t is obtained:

$$psy_t = \frac{msy_t}{r_{surv}} = \frac{(pigso_t + pigst_t)}{sowst_{t-1} \times r_{surv}} \tag{11}$$

(3) Feed price ($\omega_{21}$): Replace the feed price of sows and replacement gilts with the price of feed of fattening pigs. The main elements of concentrated feed for sows and fattening pigs are the same, and the feed price is not significantly different. Because of the availability and comparison of data, the price of feed for fattening pigs was chosen to replace the feed price for sows.

(4) Replacement gilt price ($\omega_{22}$): It is expressed by the cost of each replacement sows divided by the weight of each replacement sows.

(5) Labor price ($\omega_{23}$): The theoretical remuneration for each labor force engaged in animal husbandry production labor for a standard man day.

**The variable description of cost efficiency estimation for piglets breeding.**

1. Total cost ($C_3$): Expressed as piglet fee per piglet. Piglet fees are calculated according to the market price of similar products or actual feeding cost accounting, which can reflect the total cost of piglets before fattening. Feeding fee refers to the calculation based on the market price of similar products or the actual feeding cost, reflecting the total cost of piglets before fattening.

2. Output ($y_3$): Expressed as the number of fattening pigs that each sow can provide per year (msy).

3. Feed price ($\omega_{31}$): Replace the feed price of piglets with the price of feed of fattening pigs. The main elements of concentrated feed for piglets and fattening pigs are the same, and the feed price is not significantly different. Because of the availability and comparison of data, the price of feed for fattening pigs was chosen to replace the feed price for piglets.

4. Sow apportionment price ($\omega_{32}$): Expressed as the total cost of each sows divided by psy.

5. Labor price ($\omega_{33}$): The theoretical remuneration for each labor force engaged in animal husbandry production labor for a standard man day.

## Sample selection and data source

This paper mainly selects the 2008–2019 fattening pigs, sows, and piglets breeding panel data and uses Frontier 4.1 to measure the cost efficiency. The data come from "National Compendium of Agricultural Product Cost-benefit Data," "China Statistical Yearbook," "China Statistical Yearbook of Science and Technology," and Brick agricultural data terminals. It is worth noting that the above price-related data are all through the deflator price index to avoid the impact of multicollinearity. The descriptive statistics of specific variables are shown in Table 3. In the process of data collection, it was found that there was no relevant statistical data recorded in Xizang from 2008 to 2019 and no relevant statistical data recorded in Beijing in 2019. Considering the availability of data and the rationality and comparability of research methods, this paper takes 29 central pig-breeding provinces (autonomous regions) as the research object of regional differences in the estimated cost efficiency of fattening pigs, sows, and piglets. The 29 hog farming provinces are respectively Tianjin SHI, Shanghai SHI, Gansu Province, Qinghai Province, Ningxia Hui Autonomous Region, Xinjiang Uygur Autonomous Region, Shanxi Province, Jilin Province, Heilongjiang Province, Anhui Province, Jiangxi Province, Henan Province, Hubei Province, Hunan Province, Hebei Province, Liaoning Province, Jiangsu Province, Zhejiang Province, Fujian Province, Shandong Province, Guangdong Province, Hainan Province, Inner Mongolia Autonomous Region, Guangxi Zhuang Autonomous Region, Shaanxi Province, Sichuan Province, Chongqing SHI, Yunnan Province, Guizhou Province.

## Results and discussion

### Analysis of model estimation results

In this paper, Frontier 4.1 is used to estimate the estimated cost efficiency of fattening pigs, sows, and piglets based on the stochastic frontier cost model. The estimation results are shown

**Table 3. Descriptive statistical characteristics of main variables.**

| Variable | Explanations | Unit | Mean | Standard deviation | Minimum | Maximum |
|---|---|---|---|---|---|---|
| $C_1$ | Total cost of fattening pigs | CNY(USD)[a] | 1417.794(221.7431) | 196.1859 (30.6835) | 993.2660 (155.3468) | 2015.8570 (315.2800) |
| $C_2$ | Total cost of sows | CNY(USD) | 4341.278(678.9758) | 791.9975 (123.8684) | 2586.6000(404.5443) | 8848.4920 (1383.9040) |
| $C_3$ | Total cost of piglets | CNY(USD) | 403.9777(63.18211) | 106.5857 (16.6700) | 176.7559(27.6446) | 777.9240 (121.6673) |
| $y_1$ | The slaughter weight of fattening pigs | Kg | 116.4690 | 9.3288 | 98.3107 | 154.9701 |
| $y_2$ | Pigs weaned per Sow per Year | Head | 25.8377 | 4.9127 | 10.3012 | 39.2397 |
| $y_3$ | Market pigs per Sow per Year | Head | 22.6855 | 4.3133 | 9.0444 | 34.4525 |
| $\omega_{11}$ | Feed price | CNY/Kg (USD/Kg) | 2.4282(0.3798) | 0.4921 (0.0770) | 1.3642 (0.2134) | 5.2329 (0.8184) |
| $\omega_{12}$ | Price of piglet | CNY/Kg (USD/Kg) | 24.6172(3.8501) | 7.6423 (1.1953) | 10.5970 (1.6574) | 48.0359 (7.5128) |
| $\omega_{13}$ | Price of labor force | CNY/ Day (USD/ Day) | 54.7365(8.5608) | 18.8068(2.9414) | 21.6094(3.3797) | 100.2358 (15.6769) |
| $\omega_{22}$ | Price of replacement gilt | CNY/Kg (USD/Kg) | 11.6552(1.8229) | 2.3619 (0.3694) | 6.5480 (1.0241) | 25.1181 (3.9285) |
| $\omega_{23}$ | Daily labor price | CNY/Day (USD/Day) | 47.7061(7.4612) | 20.2367(3.1650) | 11.8000 (1.8455) | 104.3760 (16.3244) |
| $\omega_{32}$ | Sow apportionment price | CNY/Kg (USD/Kg) | 174.3332(27.2657) | 46.1747 (7.2217) | 93.2126(14.5785) | 345.8629 (54.0930) |
| envi | Environmental regulation intensity | - | 1827.7440 | 668.3808 | 390.0000 | 3829.0000 |
| tech | Technology input intensity | - | 1.4270 | 0.7771 | 0.2200 | 4.6800 |
| disea | Pig disease intensity | - | 0.0002 | 0.0010 | 0.0000 | 0.0120 |
| corn | Corn feed price | CNY/Kg (USD/Kg) | 1.8849(0.2948) | 0.2835 (0.0443) | 1.2194 (0.1907) | 2.5407 (0.3974) |
| stru | Industrial structure | - | 10.5032 | 5.1453 | 0.3000 | 30.0000 |
| ppig | Pig market price | CNY/Kg (USD/Kg) | 13.3537(2.0885) | 2.0212 (0.3161) | 9.0854(1.4210) | 18.8935 (2.9549) |

The sample size of fattening pigs, sows, and piglets is 348.

[a] This paper uses the exchange rate to convert the unit of price variables from CNY to USD. The data accounting unit in parentheses in the above table is USD.

The exchange rate is: 1CNY = 0.1564$ (Statistical time 2021-11-12 16: 13).

**Table 4. Parameters of the translog stochastic cost frontier model for fattening pigs in China.**

| Variable | Regressors Coefficient | Standard Error | T-value |
|---|---|---|---|
| constant | -22.2406*** | 1.1715 | -18.9843 |
| $lny_1$ | 10.1273*** | 0.9044 | 11.1984 |
| $lnw_{11}$ | 5.9370*** | 2.0080 | 2.9566 |
| $lnw_{12}$ | 2.5319*** | 0.8856 | 2.8589 |
| $lnw_{13}$ | -1.9082** | 0.9004 | -2.1193 |
| $lny_1^2$ | -0.7626*** | 0.1764 | -4.3232 |
| $lnw_{11}^2$ | 0.5680*** | 0.0738 | 7.7008 |
| $lnw_{12}^2$ | 0.0214 | 0.0420 | 0.5109 |
| $lnw_{13}^2$ | 0.1453*** | 0.0551 | 2.6366 |
| $lnw_{11}lnw_{12}$ | 0.1081 | 0.0933 | 1.1597 |
| $lnw_{11}lnw_{13}$ | -0.3111*** | 0.0906 | -3.4358 |
| $lnw_{12}lnw_{13}$ | 0.0298 | 0.0397 | 0.7516 |
| $lny_1lnw_{11}$ | -1.3004*** | 0.4281 | -3.0373 |
| $lny_1lnw_{12}$ | -0.5670*** | 0.1994 | -2.8441 |
| $lny_1lnw_{13}$ | 0.2067 | 0.1753 | 1.1794 |
| t | 0.0488*** | 0.0164 | 2.9717 |
| $t_2$ | -0.0030*** | 0.0010 | -3.1242 |
| constant | 23.9678** | 9.9265 | 2.4145 |
| lnenvi | -10.0968** | 4.2023 | -2.4027 |
| $(lnenvi)^2$ | 1.4043** | 0.5902 | 2.3796 |
| $(lnenvi)^3$ | -0.0650** | 0.0275 | -2.3643 |
| tech | -0.0149* | 0.0089 | -1.6806 |
| disea | -13.8559* | 7.0750 | -1.9584 |
| corn | 0.2711*** | 0.0266 | 10.1834 |
| stru | -0.0012 | 0.0013 | -0.9043 |
| $\sigma^2$ | 0.0056*** | 0.0004 | 14.1268 |
| $\gamma$ | 0.9999*** | 0.0030 | 338.1684 |
| LR | 124.546 | | |
| observations | 348 | | |

*** $p<0.01$,

** $p<0.05$,

* $p<0.1$.

Data source: Frontier's calculation results are sorted out, and the table below is the same.

in Tables 4–6. The rate of variation γ is significantly different from 0, which is the fundamental basis for the effectiveness of the stochastic frontier cost model. In the estimation results of the cost function parameters, the rate of variation γ of fattening pigs, sows, and piglets is close to 1, indicating that the random error term has a much lower impact on cost efficiency than the cost inefficiency term.

In the estimation results of cost function parameters of fattening pigs (Table 4), the variation rate γ = 0.99, indicating that the impact of random error on cost efficiency is far less than that of cost inefficiency. Under the null hypothesis that γ is equal to 0, the constraint condition is 9. The one-sided likelihood ratio test statistic LR of γ is equal to 124.55, which is greater than the mixed chi-square critical values (16.27 and 20.97) with significance levels of 0.05 and 0.01. Therefore, the null hypothesis of γ is rejected, that is, the cost inefficiency term μ exists. This shows that there is cost inefficiency in pig production in China, and it is appropriate and

**Table 5. Parameters of the translog stochastic cost frontier model for sows in China.**

| Variable | Regressors Coefficient | Standard Error | T-value |
|---|---|---|---|
| constant | 1.3937 | 0.9610 | 1.4503 |
| $\ln y_2$ | 0.7152 | 0.8866 | 0.8066 |
| $\ln w_{21}$ | -7.0745*** | 0.9726 | -7.2737 |
| $\ln w_{22}$ | 4.8591*** | 0.9590 | 5.0666 |
| $\ln w_{23}$ | -3.5446*** | 0.8533 | -4.1541 |
| $\ln y_2^2$ | -0.8326*** | 0.1993 | -4.1778 |
| $\ln w_{21}^2$ | -3.9508*** | 0.8132 | -4.8584 |
| $\ln w_{22}^2$ | -3.2973*** | 0.7468 | -4.4155 |
| $\ln w_{23}^2$ | -0.3152*** | 0.0828 | -3.8056 |
| $\ln w_{21} \ln w_{22}$ | 7.5837*** | 0.7858 | 9.6513 |
| $\ln w_{21} \ln w_{23}$ | -0.5160 | 0.5674 | -0.9095 |
| $\ln w_{22} \ln w_{23}$ | 1.8396*** | 0.6201 | 2.9667 |
| $\ln y_2 \ln w_{21}$ | -3.8630*** | 0.6121 | -6.3111 |
| $\ln y_2 \ln w_{22}$ | 2.5872*** | 0.6650 | 3.8904 |
| $\ln y_2 \ln w_{23}$ | 0.5619* | 0.2898 | 1.9388 |
| t | 0.1125** | 0.0538 | 2.0920 |
| $t_2$ | -0.0091** | 0.0036 | -2.5576 |
| constant | 2.3491** | 0.9966 | 2.3572 |
| lnenvi | 4.9432*** | 0.9549 | 5.1767 |
| $(\text{lnenvi})^2$ | -4.9846*** | 0.2818 | -17.6872 |
| $(\text{lnenvi})^3$ | 0.4328*** | 0.0213 | 20.3644 |
| tech | -2.6266*** | 0.2250 | -11.6753 |
| disea | 0.1558 | 0.9900 | 0.1558 |
| ppig | -0.0784 | 0.0625 | -1.2537 |
| stru | -0.3134*** | 0.0322 | -9.7215 |
| $\sigma^2$ | 62.7197*** | 0.7389 | 84.8779 |
| $\gamma$ | 0.9999*** | 0.000004 | 224632.11 |
| LR | 1220.0051 | | |
| observations | 348 | | |

*** $p < 0.01$,

** $p < 0.05$,

* $p < 0.1$.

necessary to establish a stochastic frontier cost model. The above analysis results show that the cost difference of fattening pigs in provinces is mainly caused by cost inefficiency. Of the 25 estimable parameters, 20 were found to be statistically significant in the models, indicating that the model as a whole is comparatively significant. Therefore, it can be determined that the stochastic frontier cost function under this assumption is still valid.

In the estimation results of cost function parameters of sows (Table 5), the variation rate $\gamma$ = 0.99, indicating that the impact of random error on cost efficiency is far less than that of cost inefficiency. Under the null hypothesis that $\gamma$ is equal to 0, the constraint condition is 9. The one-sided likelihood ratio test statistic LR of $\gamma$ is equal to 1220.01, which is greater than the mixed chi-square critical values (16.27 and 20.97) with significance levels of 0.05 and 0.01. Therefore, the null hypothesis of $\gamma$ is rejected, that is, the cost inefficiency term μ exists. This shows that there is cost inefficiency in sows production in China, and it is appropriate and

**Table 6. Parameters of the translog stochastic cost frontier model for piglets in China.**

| Variable | Regressors Coefficient | Standard Error | T-value |
|---|---|---|---|
| constant | -8710.0229*** | 2.4953 | -3490.5504 |
| $lny_3$ | 2224.5242*** | 3.3849 | 657.1938 |
| $lnw_{31}$ | -2200.4872*** | 1.3782 | -1596.6168 |
| $lnw_{32}$ | 2438.3865*** | 5.9288 | 411.2765 |
| $lnw_{33}$ | 30.9502*** | 4.0810 | 7.5840 |
| $lny_3^2$ | -141.6284*** | 0.5039 | -281.0665 |
| $lnw_{31}^2$ | -136.8319*** | 0.6220 | -219.9831 |
| $lnw_{32}^2$ | -170.1145*** | 0.8258 | -205.9979 |
| $lnw_{33}^2$ | -0.0969* | 0.0611 | -1.5855 |
| $lnw_{31}lnw_{32}$ | 306.5212*** | 0.5887 | 520.6543 |
| $lnw_{31}lnw_{33}$ | 4.6574*** | 0.5972 | 7.7994 |
| $lnw_{32}lnw_{33}$ | -4.3944*** | 0.5617 | -7.8228 |
| $lny_3lnw_{31}$ | 280.5350*** | 0.3572 | 785.4470 |
| $lny_3lnw_{32}$ | -311.8661*** | 1.2951 | -240.8054 |
| $lny_3lnw_{33}$ | -3.7578*** | 0.5450 | -6.8951 |
| t | -0.0718** | 0.0333 | -2.1541 |
| $t^2$ | 0.0015 | 0.0016 | 0.8903 |
| constant | -2.5030** | 1.0046 | -2.4917 |
| lnenvi | 1.0309** | 0.5153 | 2.0005 |
| $(lnenvi)^2$ | -0.1541 | 0.1069 | -1.4419 |
| $(lnenvi)^3$ | 0.0073 | 0.0068 | 1.0610 |
| tech | 0.0767*** | 0.0205 | 3.7478 |
| disea | -24.9785*** | 3.4762 | -7.1855 |
| ppig | 0.0675*** | 0.0042 | 15.9473 |
| stru | 0.0019 | 0.0017 | 1.0917 |
| $\sigma^2$ | 0.0329*** | 0.0025 | 13.3482 |
| $\gamma$ | 0.9999*** | 0.0108 | 92.3917 |
| LR | 133.0207 | | |
| observations | 348 | | |

*** $p<0.01$,

** $p<0.05$,

* $p<0.1$.

necessary to establish a stochastic frontier cost model. The above analysis results show that the cost difference of sows in provinces is mainly caused by cost inefficiency. Of the 25 estimable parameters, 20 were found to be statistically significant in the models, indicating that the model as a whole is comparatively significant. Therefore, it can be determined that the stochastic frontier cost function under this assumption is still valid.

In the estimation results of cost function parameters of piglets (Table 6), the variation rate $\gamma$ = 0.99, indicating that the impact of random error on cost efficiency is far less than that of cost inefficiency. Under the null hypothesis that $\gamma$ is equal to 0, the constraint condition is 9. The one-sided likelihood ratio test statistic LR of $\gamma$ is equal to 1133.02 which is greater than the mixed chi-square critical values (16.27 and 20.97) with significance levels of 0.05 and 0.01. Therefore, the null hypothesis of $\gamma$ is rejected, that is, the cost inefficiency term $\mu$ exists. This shows that there is cost inefficiency in piglets production in China, and it is appropriate and

necessary to establish a stochastic frontier cost model. The above analysis results show that the cost difference of piglets in provinces is mainly caused by cost inefficiency. Of the 25 estimable parameters, 21 were found to be statistically significant in the models, indicating that the model as a whole is comparatively significant. Therefore, it can be determined that the stochastic frontier cost function under this assumption is still valid.

### Analysis of frequency distribution of cost efficiency estimates

**Frequency distribution of cost efficiency estimates for fattening pigs.** The cost efficiency estimates of fattening pigs in China are at a medium level, showing a ' U ' trend. In this paper, the cost efficiency estimates of fattening pigs breeding are divided into five groups to analyze the cost efficiency frequency distribution of each sample province (Table 7). From 2008 to 2013, the cost efficiency estimates of fattening pig sample provinces showed a downward trend, and the average cost efficiency decreased from 0.77 to 0.71. After 2013, the cost efficiency estimates of fattening pig sample provinces entered the rising stage, showing a ' U ' trend. Provinces with cost efficiency between 0.71 and 1 gradually increased from 2013 to 2018. In 2018, provinces with cost efficiency estimates higher than 0.71 accounted for about 93% of the country. The outbreak of African swine fever in China in 2018 caused significant losses to China's s pig industry, resulting in reduced cost efficiency of fattening pigs in 2019. The average cost efficiency of fattening pig breeding is 0.77 from 2008 to 2019, which is lower than the excellent cost efficiency 1 and is at a medium level.

**Frequency distribution of cost efficiency estimates for sows.** The cost efficiency estimates of sows in China are at a medium level, showing an inverted ' U ' trend and significant regional differences. In this paper, the cost efficiency estimates of sows breeding are divided into six groups to analyze the cost efficiency frequency distribution of each sample province (Table 8). From 2008 to 2014, The cost efficiency estimates of sows increased from 0.70 to 0.90, and entered a decline stage after 2014, showing an inverted ' U ' trend. From 2015 to 2017, the proportion of sample provinces with the cost efficiency of sows above 0.71 was about 90%. However, due to the outbreak of African swine fever, only 11% of the sample provinces with sow cost efficiency estimates above 0.71 in 2019. There are great differences in the cost efficiency of sows in various provinces of China, and the gap between the maximum and minimum cost efficiency of sows is close to 2 times. In 2019, the maximum cost efficiency estimates of sows reached 0.90, and the minimum cost efficiency estimates of sows were only 0.22, indicating that there are significant differences between different sample provinces. The average cost efficiency of sow breeding from 2008 to 2019 is 0.79, which is lower than the excellent cost efficiency 1 and is at a medium level.

**Table 7. Frequency distribution of cost efficiency estimates for fattening pigs, 2008–19.**

| Efficiency Level | Year | | | | | | | | | | | |
|---|---|---|---|---|---|---|---|---|---|---|---|---|
| | 2008 | 2009 | 2010 | 2011 | 2012 | 2013 | 2014 | 2015 | 2016 | 2017 | 2018 | 2019 |
| <0.60 | 0.00 | 0.00 | 0.00 | 0.00 | 0.03 | 0.07 | 0.00 | 0.00 | 0.00 | 0.00 | 0.00 | 0.00 |
| 0.61~0.70 | 0.10 | 0.03 | 0.07 | 0.24 | 0.52 | 0.41 | 0.35 | 0.28 | 0.24 | 0.10 | 0.07 | 0.03 |
| 0.71~0.80 | 0.62 | 0.31 | 0.62 | 0.66 | 0.38 | 0.48 | 0.55 | 0.48 | 0.45 | 0.24 | 0.17 | 0.35 |
| 0.81~0.90 | 0.24 | 0.62 | 0.31 | 0.10 | 0.07 | 0.04 | 0.10 | 0.21 | 0.31 | 0.52 | 0.48 | 0.41 |
| 0.91~1.00 | 0.04 | 0.04 | 0.00 | 0.00 | 0.00 | 0.00 | 0.00 | 0.03 | 0.00 | 0.14 | 0.28 | 0.21 |
| Mean | 0.77 | 0.81 | 0.78 | 0.73 | 0.71 | 0.71 | 0.72 | 0.76 | 0.77 | 0.82 | 0.85 | 0.83 |
| Minimum | 0.62 | 0.70 | 0.66 | 0.62 | 0.54 | 0.58 | 0.63 | 0.65 | 0.67 | 0.67 | 0.67 | 0.69 |
| Maximum | 0.98 | 0.98 | 0.87 | 0.83 | 0.84 | 0.83 | 0.87 | 0.90 | 0.90 | 0.97 | 1.00 | 0.94 |

**Table 8. Frequency distribution of cost efficiency estimates for sows, 2008–19.**

| Efficiency Level | Year | | | | | | | | | | | |
|---|---|---|---|---|---|---|---|---|---|---|---|---|
| | 2008 | 2009 | 2010 | 2011 | 2012 | 2013 | 2014 | 2015 | 2016 | 2017 | 2018 | 2019 |
| <0.50 | 0.07 | 0.00 | 0.00 | 0.00 | 0.00 | 0.00 | 0.00 | 0.00 | 0.00 | 0.00 | 0.03 | 0.07 |
| 0.51~0.60 | 0.17 | 0.14 | 0.10 | 0.03 | 0.00 | 0.00 | 0.00 | 0.04 | 0.03 | 0.07 | 0.04 | 0.34 |
| 0.61~0.70 | 0.31 | 0.14 | 0.10 | 0.04 | 0.07 | 0.03 | 0.03 | 0.00 | 0.00 | 0.14 | 0.24 | 0.48 |
| 0.71~0.80 | 0.28 | 0.41 | 0.28 | 0.17 | 0.00 | 0.07 | 0.04 | 0.14 | 0.31 | 0.38 | 0.59 | 0.04 |
| 0.81~0.90 | 0.07 | 0.17 | 0.48 | 0.52 | 0.55 | 0.38 | 0.38 | 0.41 | 0.52 | 0.31 | 0.07 | 0.04 |
| 0.91~1.00 | 0.10 | 0.14 | 0.04 | 0.24 | 0.38 | 0.52 | 0.55 | 0.41 | 0.14 | 0.10 | 0.03 | 0.03 |
| Mean | 0.70 | 0.75 | 0.78 | 0.83 | 0.87 | 0.89 | 0.90 | 0.86 | 0.82 | 0.78 | 0.71 | 0.60 |
| Minimum | 0.33 | 0.51 | 0.51 | 0.52 | 0.62 | 0.67 | 0.65 | 0.57 | 0.54 | 0.52 | 0.47 | 0.22 |
| Maximum | 0.99 | 0.99 | 0.93 | 0.95 | 0.98 | 0.99 | 0.99 | 0.99 | 0.99 | 0.93 | 0.91 | 0.90 |

**Frequency distribution of cost efficiency estimates for piglets.** The cost efficiency estimates of piglets in China are at a low level with large fluctuation and significant regional differences. In this paper, the cost efficiency estimates of piglets breeding are divided into seven groups to analyze the cost efficiency frequency distribution of each sample province (Table 9). From 2008 to 2010, the average cost efficiency of piglets increased from 0.43 to 0.72, with an average annual growth of 9.67%. From 2010 to 2019, the cost efficiency estimates of piglets showed a fluctuating downward trend, with the efficiency value falling from 0.72 to 0.47, with an average annual decline of 2.78%. In 2018, the proportion of sample provinces with the cost efficiency of piglets above 0.51 was about 93%. Due to the outbreak of African swine fever, only 38% of the sample provinces with piglets cost efficiency estimates above 0.51 in 2019. The gap between the maximum and minimum cost efficiency of piglets is maintained between 1.52 and 2.59, indicating that there are significant regional differences in the cost efficiency of piglets in China. The average cost efficiency of piglet farming is 0.53 from 2008 to 2019, which is lower than the excellent cost efficiency 1 and is at a low level.

## Analysis of cost efficiency estimates in various regions

The 29 provinces studied in this paper are divided into the eastern region, central region, western region, and northeast region according to the 'China Science Statistical Yearbook'. The eastern region includes nine provinces (cities), namely Tianjin, Hebei, Shanghai, Jiangsu, Zhejiang, Fujian, Shandong, Guangdong, and Hainan. The central region consists of six provinces

**Table 9. Frequency distribution of cost efficiency estimates for piglets, 2008–19.**

| Efficiency Level | Year | | | | | | | | | | | |
|---|---|---|---|---|---|---|---|---|---|---|---|---|
| | 2008 | 2009 | 2010 | 2011 | 2012 | 2013 | 2014 | 2015 | 2016 | 2017 | 2018 | 2019 |
| <0.30 | 0.28 | 0.00 | 0.00 | 0.14 | 0.10 | 0.00 | 0.00 | 0.03 | 0.66 | 0.31 | 0.00 | 0.31 |
| 0.41~0.50 | 0.59 | 0.24 | 0.07 | 0.45 | 0.48 | 0.28 | 0.07 | 0.03 | 0.34 | 0.52 | 0.07 | 0.31 |
| 0.51~0.60 | 0.10 | 0.28 | 0.14 | 0.34 | 0.38 | 0.48 | 0.34 | 0.62 | 0.00 | 0.14 | 0.38 | 0.31 |
| 0.61~0.70 | 0.03 | 0.28 | 0.21 | 0.07 | 0.04 | 0.21 | 0.28 | 0.24 | 0.00 | 0.03 | 0.34 | 0.04 |
| 0.71~0.80 | 0.00 | 0.17 | 0.31 | 0.00 | 0.00 | 0.03 | 0.17 | 0.04 | 0.00 | 0.00 | 0.14 | 0.03 |
| 0.81~0.90 | 0.00 | 0.03 | 0.17 | 0.00 | 0.00 | 0.00 | 0.14 | 0.04 | 0.00 | 0.00 | 0.07 | 0.00 |
| 0.91~1.00 | 0.00 | 0.00 | 0.10 | 0.00 | 0.00 | 0.00 | 0.00 | 0.00 | 0.00 | 0.00 | 0.00 | 0.00 |
| Mean | 0.43 | 0.60 | 0.72 | 0.49 | 0.48 | 0.55 | 0.64 | 0.58 | 0.38 | 0.44 | 0.63 | 0.47 |
| Minimum | 0.31 | 0.41 | 0.44 | 0.34 | 0.37 | 0.42 | 0.46 | 0.40 | 0.32 | 0.36 | 0.43 | 0.28 |
| Maximum | 0.61 | 0.81 | 1.00 | 0.62 | 0.61 | 0.73 | 0.85 | 0.81 | 0.49 | 0.61 | 0.87 | 0.73 |

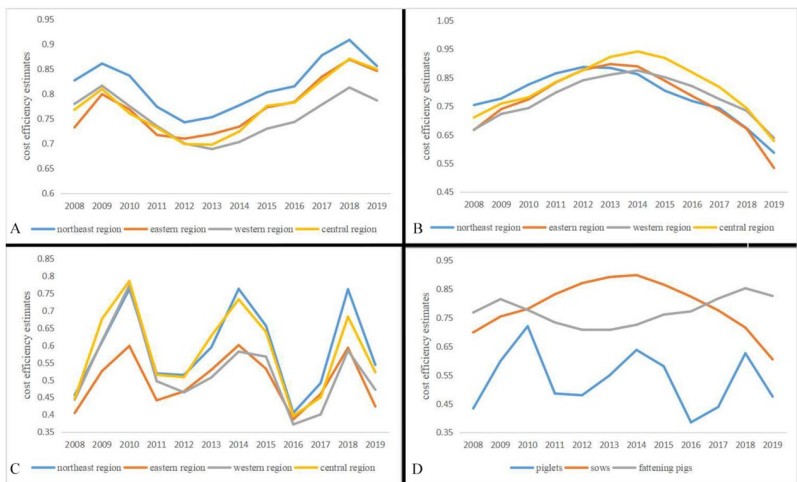

**Fig 3. The estimated cost efficiency trend in the eastern, central, western, and northeast regions (2008–2019).** A) Estimated cost efficiency estimates trend of fattening pigs (2008–2019); B) Estimated cost efficiency estimates trend of sows (2008–2019); C) Estimated cost efficiency estimates trend of piglets (2008–2019); D) The average cost efficiency trend of fattening pigs, sows and piglets (2008–2019).

(cities), namely Shanxi, Anhui, Jiangxi, Henan, Hubei, and Hunan. The western region consists of eleven provinces (cities), namely Inner Mongolia, Guangxi, Chongqing, Sichuan, Guizhou, Yunnan, Shaanxi, Gansu, Qinghai, Ningxia, and Xinjiang. The northeastern region consists of three provinces (cities) namely Liaoning, Jilin, and Heilongjiang.

**Analysis of cost efficiency estimates of fattening pigs in various regions.** The highest cost efficiency estimates of fattening pigs in the northeast region (Fig 3A). The cost efficiency estimates of fattening pigs in the northeast region are higher than that in the central, eastern, and western regions, which may be because the northeast region, as an important major grain-producing area in China, has good conditions for the development of fattening pigs breeding industry. The corn production in the northeast region accounts for more than 30% of the country, and soybean production accounts for more than 40% of the country. Grain can be processed locally, and feed is consumed locally. The feed cost of the fattening pig breeding industry is low. Before 2012, the cost efficiency of the eastern, western, and central regions had little difference. After 2012, the cost efficiency of the western region gradually lagged behind that of the eastern and central regions. It may be because the central and eastern regions are close to the densely populated main selling areas, with sufficient feed resources and low transportation costs. In addition, the labor cost of the central region is low. By introducing standardized fattening pig breeding methods, the feeding management of fattening pigs is optimized, and the breeding cost efficiency of fattening pigs is effectively improved. The cost efficiency estimates of fattening pigs in the sample provinces reached the lowest cost efficiency estimates point in 2013, which may be due to the continuous decline of pig prices in 2013, the increasing downward pressure of the feed industry, and the impact of the H7N9 influenza epidemic.

**Analysis of cost efficiency estimates of sows in various regions.** The highest cost efficiency estimates of sows in the central region (Fig 3B). The cost efficiency estimates of sows in the central region grow fastest during 2008 to 2014, and the cost efficiency estimates in the central region are higher than those in the three regions during 2013 to 2018. The main reason is that the economic development level in the central region is high, and the pressure of environmental constraints is small. Sow farmers have sufficient funds to introduce advanced

breeding technology in the central region, so the central region has a high level of cost efficiency estimates. From the perspective of the annual change trend, the decline rate of cost efficiency in the western region is relatively slow. In 2019, the cost efficiency in the western region became the highest level in the four regions, which may be due to policy influence. The provinces in the western region are the key development area and potential growth area of 'the national pig production development plan (2016 ~ 2020)'. The policy-driven makes full use of the factor resource endowment in the western region and slows down the decline of cost efficiency estimates. The decline rate of cost efficiency in the eastern region is relatively fast. The main reason may be due to the lack of self-produced feed, environmental constraints, and higher labor wages, which restrict the improvement of the cost efficiency of sows.

**Analysis of cost efficiency estimates of piglets in various regions.** The highest cost efficiency estimates of piglets in the northeast region (Fig 3C). The cost efficiency estimates of the eastern and eastern regions were consistent during 2008 to 2014. From 2014 to 2019, the cost efficiency estimates of piglets in the northeast region were gradually higher than that in the eastern, western, and central regions by resource advantages and policy support. The cost efficiency estimates of piglet breeding in the western region are low. The main reason may be that the transportation and labor costs in the western region are high, and the technological innovation is slow, which prevents the improvement of the cost efficiency of piglet breeding. Compared with the central region, the cost efficiency in the eastern region is relatively low which may be due to the high human capital and technology intensity in the eastern region, and the utilization of piglet breeding factor resources has tended to be saturated. In addition, the environmental constraints in the eastern region are strong, resulting in more input and less output increase. Compared with sows and fattening pigs, the cost efficiency estimated of piglets is the lowest (Fig 3D).

## Analysis of influencing factors of cost inefficiency

**Analysis of influencing factors of cost inefficiency of fattening pigs.** According to Frontier 4.1 estimation results (Table 10), the four influencing factors that are significantly related to the inefficiency of fattening pigs are environmental regulation intensity, technological input intensity, pig disease intensity, and corn feed price. As shown in Table 10, the first term of environmental regulation intensity is negative, and the square term is positive, and the cubic term is negative. This shows that the relationship between environmental regulation intensity and cost efficiency of fattening pigs is an ' N ' curve. The environmental regulation intensity

**Table 10. The estimation results of influencing factors of cost inefficiency model for fattening pigs.**

| Variable | Regressors Coefficient | Standard Error | T-value |
|---|---|---|---|
| constant | 23.9678** | 9.9265 | 2.4145 |
| $lnenvi_{it}$ | -10.0968** | 4.2023 | -2.4027 |
| $(lnenvi_{it})^2$ | 1.4043** | 0.5902 | 2.3796 |
| $(lnenvi_{it})^3$ | -0.0650** | 0.0275 | -2.3643 |
| $tech_{it}$ | -0.0149* | 0.0089 | -1.6806 |
| $disea_{it}$ | -13.8559** | 7.0750 | -1.9584 |
| $corn_{it}$ | 0.2711*** | 0.0266 | 10.1834 |
| $stru_{it}$ | -0.0012 | 0.0013 | -0.9043 |

*** $p < 0.01$,

** $p < 0.05$,

* $p < 0.1$.

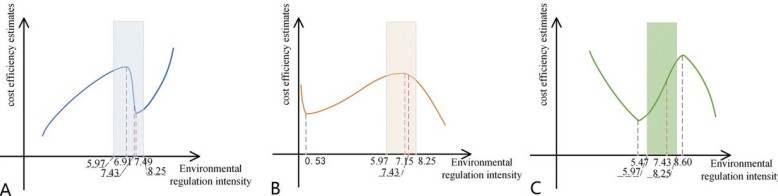

**Fig 4. Nonlinear relationship between environmental regulation intensity and cost efficiency estimates.** A) the relationship between environmental regulation intensity and cost efficiency of fattening pigs; B) the relationship between environmental regulation intensity and cost efficiency of sows; C) the relationship between environmental regulation intensity and cost efficiency of piglets. A)-C) is only a rough figure representing the relationship between environmental regulation intensity and cost efficiency and is not an accurate functional relationship figure. A)-C) can only reflect the promotion or inhibition of environmental regulation intensity on cost efficiency, and provide an inflection point for the analysis of the paper.

coefficients passed the significance test at the 5% confidence level, which showed that the model fitting was good and had strong explanatory power. By calculation, the two inflection points of environmental regulation are 6.91 and 7.49 (Fig 4A). The average intensity of environmental regulation policies in China's hog industry is 7.43 (Table B in S1 Table), which is right of the first inflection point and left of the second inflection point, indicating that the impact of environmental regulation policy on the cost efficiency of fattening pigs is in the 'N' decline stage. Technological input intensity at the level of 10% has a significant negative impact on the cost inefficiency of fattening pigs, indicating that increasing the level of technical input is conducive to improving the cost efficiency of fattening pigs. The pig disease intensity at the 5% test level has a significant negative impact on the cost inefficiency of fattening pigs. The main reason is that the outbreak of the disease will accelerate the scaling process of the hog industry. With the increase of the scale of breeding, the efficiency increases, far more than the decrease of cost efficiency caused by the outbreak of the disease. The corn feed price was significantly positive at the 1% level. Reducing the corn feed price effectively reduced the production cost, which was beneficial to improve the cost efficiency of fattening pigs.

**Analysis of influencing factors of cost inefficiency of sows.** According to Frontier4.1 estimation results (Table 11), environmental regulation intensity, technology input intensity, and industrial structure are the three main factors significantly related to the sow cost inefficiency. As shown in Table 11, the first term of environmental regulation intensity is positive, and the

**Table 11. The estimation results of influencing factors of cost inefficiency model for sows.**

| Variable | Regressors Coefficient | Standard Error | T-value |
|---|---|---|---|
| constant | 2.3491** | 0.9966 | 2.3572 |
| $lnenvi_{it}$ | 4.9432*** | 0.9549 | 5.1767 |
| $(lnenvi_{it})^2$ | -4.9846*** | 0.2818 | -17.6872 |
| $(lnenvi_{it})^3$ | 0.4328*** | 0.0213 | 20.3644 |
| $tech_{it}$ | -2.6266*** | 0.2250 | -11.6753 |
| $disea_{it}$ | 0.1558 | 0.9900 | 0.1558 |
| $lnpig_{it}$ | -0.0784 | 0.0625 | -1.2537 |
| $stru_{it}$ | -0.3134*** | 0.0322 | -9.7215 |

*** $p < 0.01$,

** $p < 0.05$,

* $p < 0.1$.

**Table 12. The estimation results of influencing factors of cost inefficiency model for piglets.**

| Variable | Regressors Coefficient | Standard Error | T-value |
|---|---|---|---|
| constant | -2.5030** | 1.0046 | -2.4917 |
| $lnenvi_{it}$ | 1.0309** | 0.5153 | 2.0005 |
| $(lnenvi_{it})^2$ | -0.1541 | 0.1069 | -1.4419 |
| $(lnenvi_{it})^3$ | 0.0073 | 0.0068 | 1.0610 |
| $tech_{it}$ | 0.0767*** | 0.0205 | 3.7478 |
| $disea_{it}$ | -24.9785*** | 3.4762 | -7.1855 |
| $lnpig_{it}$ | 0.0675*** | 0.0042 | 15.9473 |
| $stru_{it}$ | 0.0019 | 0.0017 | 1.0917 |

*** $p < 0.01$,

** $p < 0.05$,

* $p < 0.1$.

square term is negative, and the cubic term is positive. This shows that the relationship between environmental regulation intensity and cost efficiency of sows is an inverted ' N ' curve. The environmental regulation intensity coefficients passed the significance test at the 1% confidence level, which showed that the model fitting was good and had strong explanatory power. By calculation, the two inflection points of environmental regulation are 0.53 and 7.15 (Fig 4B). The average intensity of environmental regulation policies in China's hog industry is 7.43 (Table B in S2 Table), which is on the right of the second inflection point, indicating that the impact of environmental regulation policy on the cost efficiency of sows is in the inverted ' N ' decline stage. Technology input intensity has a significantly negative impact on the inefficiency of sow cost at the 1% test level, indicating that improving the level of technological input is conducive to improving the cost efficiency of sow breeding. The industrial structure has a significant negative impact on the cost efficiency estimates of sows at the 1% test level. The main reason is that the areas with high agricultural resources endowment can process food locally, consume feed locally, and have low feed cost, which is beneficial to improve the cost efficiency of sows.

**Analysis of influencing factors of cost inefficiency of piglets.** According to Frontier4.1 estimation results (Table 12), environmental regulation intensity, technology input intensity, pig disease intensity, and pig market price are the four main factors significantly related to the cost inefficiency of piglets. As shown in Table 12, the first term of environmental regulation intensity is positive passed the significance test at the 1% confidence level. With the increase of environmental regulation intensity, the cost efficiency of piglets will show an inverted "N" trend. By calculation, the two inflection points of environmental regulation are 5.47 and 8.60 (Fig 4C). The average intensity of environmental regulation policies in China's hog industry is 7.43 (Table B in S3 Table), which is right of the first inflection point and left of the second inflection point, indicating that the impact of environmental regulation policy on the cost efficiency of piglets is in the inverted ' N ' rising stage. The technology input intensity has a significant positive effect on the cost inefficiency of piglets at 1% test level, indicating that reducing the technical input level is beneficial to improve the cost efficiency of piglet breeding. The main reason is that the technical input in piglet breeding will greatly increase the breeding cost and hinder the improvement of cost efficiency estimates. The pig disease intensity has a significantly negative impact on the cost inefficiency of piglets at the test level of 1%, indicating that when the epidemic broke out, farmers have improved their management level, which will increase the breeding cost in the short term. However, in the long term, the outbreak of the epidemic also forces industrial upgrading, increasing the overall cost efficiency. Pig market

price is significantly positive at 1% level, indicating that the pig market price increases, farmers will blindly expand production, is not conducive to the coordination of factor allocation, inhibit the continuous improvement of cost efficiency estimates.

## Conclusions

Nowadays, China's hog industry is in a critical period of stable production and supply. Facing the dual constraints of resources and environment, how to reduce production costs and achieve efficiency gains has become an unavoidable problem for the sustainable development of the hog industry. Based on the provincial panel data during 2008 to 2019, this paper uses the method of the translog stochastic frontier cost function to estimate the cost efficiency of fattening pigs, piglets, and sows in China, focusing on the impact of environmental regulation on cost efficiency.

The main conclusions of this paper are as follows. First, the estimated cost efficiency of fattening pigs, sows, and piglets in China is relatively low, and there are different degrees of cost inefficiency. Based on these results, by operating at the efficient frontier the sample fattening pigs producers would be able to reduce their production cost by 23%, the sample sows producers would be able to reduce their production cost by 21%, the sample piglets producers would be able to reduce their production cost by 47%. Second, for fattening pigs, further improving the intensity of environmental regulation and crossing the second inflection point of the ' N ' curve can achieve the dual goals of environmental governance and cost reduction. Increasing the intensity of technology input can also achieve efficiency gains. Third, for sows, appropriately reducing the intensity of environmental regulation can avoid cost efficiency estimates losses. Farmers improve the intensity of technology input and transfer to areas with high agricultural factor resource endowment can achieve cost efficiency gains. Finally, for piglets, environmental regulation does not form an effective incentive for cost efficiency; farmers to reduce the intensity of technology input, establish price early warning mechanism can avoid cost efficiency estimates loss.

The main findings of this paper are as follows. First, to avoid the loss of efficiency, it is necessary for farmers to actively adapt to the adjustment of environmental regulation policies through organic fertilizer processing and biogas project construction. Second, making full use of the regional advantages. Northeast region, as the potential growth area of the ' National Pig Production Development Plan (2016 ~ 2020) ', the hog industry development emphasis should be on fattening pigs and piglet breeding. The central region belongs to the moderate development area and constraint development area of ' National Pig Production Development Plan (2016 ~ 2020) '. The industrial development space is limited, and the focus of the pig industry development should be on sows. Third, to achieve the gain of cost efficiency, it is necessary for fattening pigs and sows breeding enterprises to increase the level of technological R&D investment, strengthen the dissemination and training of technological achievements, and improve the professional quality of breeding practitioners. Finally, the hog breeding industry should establish a price early warning mechanism, monitor the price information and supply and demand status of each link of the industrial chain in all aspects. In addition, timely disclose information to avoid panic among farmers caused by external shocks such as major animal epidemics. Farmers can make full use of ' insurance + futures ' and other emerging financial instruments to avoid market risks and efficiency losses.

## Supporting information

**S1 Table. Input and output indicators for fattening pigs in China for 2008–2019.**
(XLSX)

**S2 Table. Input and output indicators for sows in China for 2008–2019.**
(XLSX)

**S3 Table. Input and output indicators for piglets in China for 2008–2019.**
(XLSX)

## Acknowledgments

We would like to thank the editors and two reviewers whose comments helped to greatly improve this manuscript.

## Author Contributions

**Conceptualization:** Gangyi Wang, Chang'e Zhao, Yuzhuo Shen.

**Data curation:** Chang'e Zhao, Yuzhuo Shen.

**Formal analysis:** Gangyi Wang, Chang'e Zhao.

**Funding acquisition:** Gangyi Wang.

**Methodology:** Gangyi Wang, Chang'e Zhao, Yuzhuo Shen.

**Project administration:** Gangyi Wang.

**Resources:** Gangyi Wang, Chang'e Zhao, Yuzhuo Shen, Ni Yin.

**Supervision:** Gangyi Wang.

**Validation:** Gangyi Wang, Chang'e Zhao.

**Visualization:** Gangyi Wang, Chang'e Zhao, Ni Yin.

**Writing – original draft:** Gangyi Wang, Chang'e Zhao.

**Writing – review & editing:** Gangyi Wang, Chang'e Zhao.

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
