## [Decision Letter · Decision Letter 0]

12 Oct 2021

PONE-D-21-22179Estimation of Cost Efficiency of fattening pigs, sows, and piglets using SFA approach analysis: evidence from ChinaPLOS ONE

Dear Dr. chang'e zhao,

Thank you for submitting your manuscript to PLOS ONE. After careful consideration, we feel that it has merit but does not fully meet PLOS ONE’s publication criteria as it currently stands. Therefore, we invite you to submit a revised version of the manuscript that addresses the points raised during the review process.

Please submit your revised manuscript by Nov 26 2021 11:59PM. If you will need more time than this to complete your revisions, please reply to this message or contact the journal office at plosone@plos.org. Please include the following items when submitting your revised manuscript:A rebuttal letter that responds to each point raised by the academic editor and reviewer(s). You should upload this letter as a separate file labeled 'Response to Reviewers'.A marked-up copy of your manuscript that highlights changes made to the original version. You should upload this as a separate file labeled 'Revised Manuscript with Track Changes'.An unmarked version of your revised paper without tracked changes. You should upload this as a separate file labeled 'Manuscript'.

We look forward to receiving your revised manuscript.

Kind regards,

Carlos Alberto Zúniga-González, Ph.D

Academic Editor

PLOS ONE

Journal Requirements:

National Natural Science Foundation of China (71673273); Heilongjiang Provincial Natural Science Foundation of China (LH2019G002); Chongqing Municipal Education Commission Management and Model Innovation Project (KJQN201901350)

3. We note that Figure 1, 3, 5 and 7 in your submission contain [map/satellite] images which may be copyrighted. All PLOS content is published under the Creative Commons Attribution License (CC BY 4.0), which means that the manuscript, images, and Supporting Information files will be freely available online, and any third party is permitted to access, download, copy, distribute, and use these materials in any way, even commercially, with proper attribution. For these reasons, we cannot publish previously copyrighted maps or satellite images created using proprietary data, such as Google software (Google Maps, Street View, and Earth). For more information, see our copyright guidelines: http://journals.plos.org/plosone/s/licenses-and-copyright.

a. You may seek permission from the original copyright holder of Figure 1, 3, 5 and 7 to publish the content specifically under the CC BY 4.0 license.  

Additional Editor Comments:

Dear authors, in order to improve the quality of your manuscript, I suggest making the improvements indicated by the reviewers, in the same way I ask you to consider the suggested references.

References

share some literature that you can use to reinforce some aspects of the DEA approach and its use with the Tobbit model:

1) González, C. A. Z. (2011). Technical efficiency of organic fertilizer in small farms of Nicaragua: 1998-2005. African Journal of Business Management, 5(3), 967-973. Available from publons.com/p/11272633/

2) Dios-Palomares, R. (2015). 7. Analysis of the Efficiency of Farming Systems in Latin America and the Caribbean Considering Environmental Issues. Revista Cientifica-Facultad de Ciencias Veterinarias, 25(1). Available in publons.com/p/3106827/

3) Zuniga González, C. (2020). Total factor productivity growth in agriculture: Malmquist index analysis of 14 countries, 1979-2008 . REICE: Revista Electrónica De Investigación En Ciencias Económicas, 8(16), 68-97. https://doi.org/10.5377/reice.v8i16.10661 Available from publons.com/p/36247914/

4) Blanco-Orozco, N., Arce-Díaz, E., & Zúñiga-Gonzáles, C. (2015). Integral assessment (financial, economic, social, environmental and productivity) of using bagasse and fossil fuels in power generation in Nicaragua. Revista Tecnología en Marcha, 28(4), 94-107. Available from publons.com/p/32281799/

Reviewers' comments:

Reviewer's Responses to Questions

**Comments to the Author**

1. Is the manuscript technically sound, and do the data support the conclusions?

Reviewer #1: No

Reviewer #2: Partly

2. Has the statistical analysis been performed appropriately and rigorously? 

Reviewer #1: No

Reviewer #2: Yes

3. Have the authors made all data underlying the findings in their manuscript fully available?

Reviewer #1: No

Reviewer #2: Yes

4. Is the manuscript presented in an intelligible fashion and written in standard English?

Reviewer #1: No

Reviewer #2: Yes

5. Review Comments to the Author

Reviewer #1: The article under review aims at measuring cost efficiency of three different pig farming systems in China covering the period 2008-2018 and analyse the influence of environmental regulation policies in explaining the calculated cost efficiency.

Overall, the paper suffers from a major methodological flaw that precludes its publication in its current form. The present paper uses a two-stage equations to (1) estimate the cost-efficiency through SFA and (2) to analyse the efficiency determinants. This approach is considered to be not the appropriate method in the SFA framework as the error terms of the two equations could be correlated.

I suggest using one of the following approaches:

• One stage estimation through SFA: The determinants of the inefficiency should be explicitly introduced in model. See (Coelli & Battese, 1996; Kumbhakar et al., 1991; Tzouvelekas et al., 2001)

• Two-stage estimation through DEA: you can follow the same approach as your original analysis, with the estimation of DEA efficiency scores and then you perform a Tobit model to analyse the effects of contextual variables. It is worth mentioning that a censored Tobit model might be inappropriate because of serial correlation, therefore, you could apply a more robust estimation method such as the one proposed by Simar & Wilson (2007).

References

Coelli, T., & Battese, G. (1996). Identification of factors which influence the technical inefficiency of Indian farmers. Australian Journal of Agricultural and Resource Economics, 40(2), 103–128. https://doi.org/10.1111/j.1467-8489.1996.tb00558.x

Kumbhakar, S. C., Ghosh, S., & McGuckin, J. T. (1991). A Generalized Production Frontier Approach for Estimating Determinants of Inefficiency in U.S. Dairy Farms. Journal of Business & Economic Statistics, 9(3), 279. https://doi.org/10.2307/1391292

Simar, L., & Wilson, P. W. (2007). Estimation and inference in two-stage, semi-parametric models of production processes. Journal of Econometrics, 136(1), 31–64. https://doi.org/10.1016/j.jeconom.2005.07.009

Tzouvelekas, V., Pantzios, C. J., & Fotopoulos, C. (2001). Technical efficiency of alternative farming systems: The case of Greek organic and conventional olive-growing farms. Food Policy, 26(6), 549–569. https://doi.org/10.1016/S0306-9192(01)00007-0

Reviewer #2: Review of the manuscript PONE-D-21-22179 by Gangyi Wang et al.

The manuscript describes retrospective analysis of data originating from official data sources like National Compendium of Agricultural Product Cost-benefit Data, China Statistical Yearbook, China Statistical Yearbook of Science and Technology, and Brick agricultural data terminals. The Authors have chosen 23 provinces representing intensive swine farming area. The data from decade (2008-2018) were analyzed using two standard models to receive information about cost efficiency in macroeconomic scale.

The manuscript seem to be interesting. The analysis is of practical importance, including potential to use models for prediction of cost and production efficiency after relevant changes of production environment, especially administrative burdens. That is why in my opinion the manuscript meets the requirements of PLOS One in terms of the scope of the Journal. However, some amendments are needed before acceptance.

Introduction is well written and characterize clearly the practical importance of research. However, there is no clearly defined aim of the study at the end of Introduction. The last paragraph seem to be more M&M summary, so the aim of the study is defined only as one general sentence at the beginning of Abstract. It is not enough, and clear definition of the aim of the study must be add. Additionally I would suggest to add information about more international currency (e.g. USD), because the value of yuan is difficult to interpret for international reader.

M&M

The study is based on retrospective analysis of macroscale data, that is why the most important is to define clearly variables to analyze and models. The models are defined, however variables only generally in the text, and a little more detailed in table 3. This table must be corrected, because variables in table poorly correspond to the text of M&M, and thus the table is difficult to interpret. First of all, variables in the table should be named to be interpretable reading only table (without text). And second, variables should be characterized also with units (is this only cost in currency, or the other units were used). In my opinion, it would be also interesting to know production parameters to compare them with costs. The Authors base on statistical year books, so I am pretty sure, that such data is also possible to obtain. It is important because e.g. feed intake and FCR influence costs of production significantly.

Results are described very detailed, and after corrections in M&M should be OK.

Discussion is very general and seem to be more reediting of results than critical interpretation. The specificity of data analysis suggests, that maybe it would be better to compose manuscript without separate discussion, and change it to Results and discussion as one section. Anyway, this section should and with conclusion. There is no such separate section, and the manuscript seem to have no clear end, and no suggestion for practice. This must be corrected and completed.

To summarize, in my opinion the manuscript could be interesting, but must be improved. Minor language revision is also required.

6. PLOS authors have the option to publish the peer review history of their article (what does this mean?). If published, this will include your full peer review and any attached files.

Reviewer #1: No

Reviewer #2: No

---

## [Author Response · Author response to Decision Letter 0]

22 Nov 2021

Thank you for your valuable feedback of our manuscript entitled " Estimation of cost efficiency of fattening pigs, sows, and piglets using SFA approach analysis: evidence from China ". We have studied comments carefully. (Specific revision details are given in “Response to Reviewers”)

---

## [Editor Report · Decision Letter 1]

26 Nov 2021

Estimation of cost efficiency of fattening pigs, sows, and piglets using SFA approach analysis: evidence from China

PONE-D-21-22179R1

Dear Dr. chang'e zhao,

We’re pleased to inform you that your manuscript has been judged scientifically suitable for publication and will be formally accepted for publication once it meets all outstanding technical requirements.

Kind regards,

Carlos Alberto Zúniga-González, Ph.D

Academic Editor

PLOS ONE

Additional Editor Comments (optional):

Dear authors, I sincerely congratulate you for the effort made to improve the manuscript, I see that you have made a good effort and followed the authors' suggestions, although I do not see the suggested references incorporated.
---

## [Editor Report · Acceptance letter]

3 Dec 2021

PONE-D-21-22179R1 

Estimation of cost efficiency of fattening pigs, sows, and piglets using SFA approach analysis: evidence from China 

Dear Dr. Zhao:

I'm pleased to inform you that your manuscript has been deemed suitable for publication in PLOS ONE. Congratulations! Your manuscript is now with our production department. 

Kind regards, 

on behalf of

Dr. Prof. Carlos Alberto Zúniga-González 

Academic Editor

PLOS ONE